# Fixed vs adjusted-dose benznidazole for adults with chronic Chagas disease without cardiomyopathy: A systematic review and meta-analysis

**Agustín Ciapponi**[1,2]*, **Fabiana Barreira**[3], **Lucas Perelli**[1], **Ariel Bardach**[1,2], **Joaquim Gascón**[4], **Israel Molina**[5], **Carlos Morillo**[6], **Nilda Prado**[7], **Adelina Riarte**[7], **Faustino Torrico**[8], **Isabela Ribeiro**[9], **Juan Carlos Villar**[10], **Sergio Sosa-Estani**[2,3]*

**1** Centro Cochrane Argentino-Instituto de Efectividad Clínica y Sanitaria (IECS-CONICET), Buenos Aires, Argentina, **2** Centro de Investigaciones Epidemiológicas y Salud Pública (CIESP-IECS). CONICET, Buenos Aires, Argentina, **3** Drugs for Neglected Diseases *initiative* (DND*i*), Río de Janeiro, Brazil, **4** Hospital Clínic de Barcelona, Barcelona, España, **5** Hospital Universitari Vall d'Hebron Research Institute, Barcelona, España, **6** McMaster University, Population Health Research Institute, Hamilton, Canada, **7** Instituto Nacional de Parasitología Dr. M Fatala Chaben, Buenos Aires, Argentina, **8** Universidad Mayor de San Simón, Cochabamba, Bolivia, **9** Drugs for Neglected Diseases *initiative* (DND*i*), Geneva, Switzerland, **10** Grupo de Cardiología Preventiva, Facultad de Ciencias de la Salud, Universidad Autónoma de Bucaramanga, Bucaramanga, Colombia

* aciapponi@iecs.org.ar (AC); ssosa@dndi.org (SS-E)

**Data Availability Statement:** All data is presented in the supporting information of the manuscript.

## Abstract

Chagas disease is a neglected disease that remains a public health threat, particularly in Latin America. The most important treatment options are nitroimidazole derivatives, such as nifurtimox and benznidazole (BZN). Some studies suggest that for adults seropositive to *T. cruzi* but without clinically evident chronic Chagas cardiomyopathy (CCC), a simple fixed-dose scheme of BZN could be equivalent to a weight-adjusted dose. We compared the efficacy and safety of a fixed dose of BZN with an adjusted dose for *T. cruzi* seropositive adults without CCC. We used the Cochrane methods, and reported according to the PRISMA statement. We included randomized controlled trials (RCTs) allocating participants to fixed and/or adjusted doses of BZN for *T. cruzi* seropositive adults without CCC. We searched (December 2019) Cochrane, MEDLINE, EMBASE, LILACS, Clinicaltrials.gov, and International Clinical Trials Registry Platform (ICTRP), and contacted Chagas experts. Selection, data extraction, and risk of bias assessment, using the Cochrane tool, were performed independently by pairs of reviewers. Discrepancies were solved by consensus within the team. Primary outcomes were parasite-related outcomes and efficacy or patient-related safety outcomes. We conducted a meta-analysis using RevMan 5.3 software and used GRADE summary of finding tables to present the certainty of evidence by outcome. We identified 655 records through our search strategy and 10 studies (four of them ongoing) met our inclusion criteria. We did not find any study directly comparing fixed vs adjusted doses of BZN, however, some outcomes allowed subgroup comparisons between fixed and adjusted doses of BZN against placebo. Moderate-certainty evidence suggests no important subgroup differences for positive PCR at one year and for three safety outcomes (drug

**Funding:** This study was funded by an independent grant from Drugs for Neglected Diseases initiative (DNDi) that had no role in the data collection and analysis, decision to publish, or preparation of the manuscript. Two out of the 13 authors are from DNDi.

**Competing interests:** The authors have declared that no competing interests exist.

discontinuation, peripheral neuropathy, and mild rash). The same effect was observed for any serious adverse events (low-certainty evidence). All subgroups showed similar effects ($I^2$ 0% for all these subgroup comparisons but 32% for peripheral neuropathy), supporting the equivalence of BZN schemes.

We conclude that there is no direct evidence comparing fixed and adjusted doses of BZN. Based on low to very low certainty of evidence for critical clinical outcomes and moderate certainty of evidence for important outcomes, fixed and adjusted doses may be equivalent in terms of safety and efficacy. An individual patient data network meta-analysis could better address this issue.

## Author summary

Chagas disease is a major public health problem that requires, among other control interventions, an optimal trypanocidal therapy that achieves the best possible compliance to cure active infection, mainly in children and young populations, women before they become pregnant to prevent congenital transmission, and chronic populations who are currently not being treated and with a risk of progression to cardiomyopathy. Some studies suggest that a simple fixed-dose scheme of benznidazole could be equivalent to the dose adjusted by weight for the treatment of adults seropositive to *T. cruzi* without clinically evident chronic Chagas cardiomyopathy. To confirm or reject this potential equivalence of schemes, we conducted a rigorous systematic review and meta-analysis of randomized controlled trials by reviewing and analyzing the totality of available literature on the subject. Although we did not find direct evidence addressing this question, it appears that an adjusted dose is probably equivalent in terms of important safety and efficacy outcomes, while the effect on critical outcomes is uncertain. Since we did not find any ongoing study comparing fixed versus adjusted doses of benznidazole, we are conducting an individual patient data network meta-analysis to address this question.

## Introduction

Chagas disease (CD), also known as human American trypanosomiasis, is a condition resulting from infection by the parasite *T. cruzi*. Chagas disease remains a major public health problem; between five and 18 million people are currently infected and the disease is estimated to cause more than 10,000 deaths annually[1]. Globally, the annual burden is $627.5 million in health-care costs[2], and 232,000 to 806,170 disability-adjusted life years (DALYs)[3]. The Latin American region bears most of the burden of Chagas disease, accounting for at least 206,000[2] to 662,000 DALYs lost[4]. A study investigating the economic value of a therapeutic Chagas vaccine found that when administering standard of care benznidazole (BZN) to 1000 indeterminate patients, 148 discontinued treatment and 219 progressed to chronic disease, resulting in 119 Chagas-related deaths and 2293 DALYs, costing $18.9 million in lifetime societal costs[5]. Population migration dynamics combined with the increased risk of mothers infecting their unborn children and the increased risk of infection from blood or solid-organ donations, means that CD has become a global problem[6]. The number of infected individuals has been estimated at 300,000 in the USA,[7] and 80,000 in Europe[8]. Primary acute *T. cruzi* infection is seldom clinically evident, given its lack of defining features. CD is often asymptomatic or resembles a common viral illness, although more serious outcomes such as

myocarditis or meningoencephalitis are possible. During this period, *T. cruzi* trypomastigotes are directly observable in the bloodstream. After this comes an indeterminate chronic phase, during which *T. cruzi* lodges in organ tissue in amastigote form, inducing a specific immune response. While most remain asymptomatic, 30–40% of patients progress to an advanced disease stage, usually years to decades after the initial infection. The advanced chronic phase frequently involves damage to the conduction system of the heart and the myocardium, which can result in heart failure and sudden death. In the Americas, myocarditis secondary to CD is the most common form of nonischemic cardiomyopathy[9]. In other cases, CD produces gastrointestinal disorders (especially megaesophagus and megacolon), or disorders of the central or peripheral nervous system, particularly in immunocompromised patients. Serology is used to confirm a diagnosis of chronic *T. cruzi* infection and polymerase chain reaction (PCR) contributes to that diagnosis. For years, host-based control was considered a difficult goal to achieve and consequently, in the 1990s, public health authority efforts were focused on primary prevention, *Triatoma infestans*-based control, and control of blood donors to prevent infection of individuals at risk[10]. In recent years, significant progress has been made in the fight against triatomines, which, added to the controls implemented by blood banks, has drastically reduced *T. cruzi* infections by vectors and transfusions. Interest in host-based control, that is treating chronically infected individuals with trypanocidal therapy, has increased[11, 12]. Additionally, the focus on vector-based control has left the already infected population without interventions that are potentially preventive of CCC[13]. Two nitroimidazolic derivatives, BZN and nifurtimox, are the only approved trypanocidal options currently used, with no important differences in their relative efficacy, adverse effects (AEs) and cost[14]. The usual recommended dose of BZN is 5 to 7 mg/kg/day orally (5–10 mg/kg for children up to 12 years old) divided into two or three times daily, for 60 days for adults. The most frequently reported side effects are skin reactions and neuropathy, which commonly result in interruption of treatment[15].

Recently, several studies[16–18] suggested that the use of a simpler fixed dose of BZN may be equivalent to an adjusted dose in terms of effectiveness, simplifying its administration and enhancing compliance. In order to compare the efficacy and safety of both schemes for *T. cruzi* seropositive adults without CCC, we have systematically searched and extracted data from eligible studies comparing relevant clinical, parasitological, and biochemical outcomes for seropositive adults exposed to fixed and/or adjusted doses of BZN.

## Methods

We conducted a systematic review and meta-analysis following Cochrane methods,[19] and the PRISMA statement for reporting[20, 21]. The study protocol was registered in PROSPERO (CRD42019120905).

### Eligibility criteria

Randomized controlled trials (RCTs) allocating adults with asymptomatic chronic Chagas disease to fixed and/or adjusted doses BZN vs placebo or other trypanocidal treatments were included. RCTs had to include people with chronic *T. cruzi* infection, diagnosed with positive serology by at least two of the following techniques: ELISA, indirect hemagglutination (IHA), or indirect immunofluorescence (IIF), mainly without clinically evident (i.e. symptomatic) CCC. Important safety and efficacy outcomes, including proxies as positive serology or PCR, any adverse events (AEs) and serious adverse events (SAEs) and critical patient (clinical) related outcomes, such as all-cause mortality or significant progression of CCC, were analyzed.

## Search strategy

We searched during December 2019 the following databases: Cochrane Database of Systematic Reviews (CDSR), Cochrane Central Register of Controlled Trials (CENTRAL), Database of Abstracts of Reviews of Effects (DARE), MEDLINE, EMBASE, LILACS, Clinicaltrials.gov, and the WHO International Clinical Trials Registry Platform (ICTRP).

The basic search strategy included the following terms: (Chagas Disease [Mesh] OR Chagas [tiab] OR Trypanosom*[tiab] OR Cruzi[tiab] OR T.Cruzi[tiab]) AND (Benznidazole[Supplementary Concept] OR benznidazol*[tiab] OR Radanil[tiab] OR Rochagan[tiab] OR N-bencil-2-acetamide[tiab]).

The search strategy was adapted to each database (See S1 Text).

No language limitations or publication date restrictions were applied. For studies with multiple publications, we decided how to best use the data on a case-by-case basis through discussion with the principal investigators.

Additional searches included a Google search (the first 100 hits, in order of relevance, when typing Chagas benznidazol), handsearching of reference lists of systematic reviews and eligible studies retrieved with the electronic search, and verbal feedback from experts in the field.

## Screening and data extraction

Selection, data extraction, and risk of bias assessment were performed independently by pairs of reviewers from the research team. Discrepancies were solved by consensus within the team. All the study selection phases were completed using COVIDENCE, a web-based platform designed for the systematic review process. Authors of articles were contacted when necessary to obtain missing or supplementary information.

A pre-designed general data extraction form was used after pilot testing.

We extracted the source of study report, study location and setting, population, disease definition of chronic *T. cruzi* infection, diagnostic tests used (number and type of laboratory tests used), quality control measures, BZN and other treatment schedules, and follow-up measures.

## Risk of bias (quality) assessment

Pairs of independent reviewers assessed the risk of bias (quality) using the Cochrane Collaboration tool[22]. See details in S2 Text. Discrepancies were solved by consensus within the team.

## Data synthesis

We conducted a traditional aggregate meta-analysis by using the Review Manager 5.3 software package[23]. Pooled effect estimates and their 95% confidence intervals (CI), when appropriate, were generated using a random-effects model. We reported risk ratios (RR) for dichotomous outcomes (e.g. positive serology after treatment), the Peto odds ratio (OR) for very infrequent outcomes, and the mean difference (MD) for continuous data such as antibody titers. For dichotomous data, we used the Mantel-Haenszel method, and for continuous data, we used the inverse variance method.

We described statistical heterogeneity of intervention effects by calculating the $I^2$ statistic and we interpreted 0–30% as not important, 30–60% as moderate heterogeneity, and more than 60% as substantial heterogeneity. Since we assume that clinical heterogeneity is very likely to impact our review results, given the nature of the interventions included, we primarily reported the random-effects model results, however, we also applied the fixed effect model as a sensitivity analysis. We calculated all overall effects using inverse variance methods.

Where necessary, we contacted the corresponding authors of included studies up to three times to supply any unreported data.

We planned subgroup analyses (see protocol) by age of participants at time of treatment (young adults vs. older population), type of serological test (conventional serology vs. non-conventional serology), time of treatment and testing (less than ten years vs. equal or more than ten years), and region where the patient was infected (Central vs. South America). We expected, *ex ante*, to find an earlier and higher rate of seroconversion in Central America due to the presence of different parasite lineages, i.e., *T. cruzi* type I predominating in Central America and *T. cruzi* type Non-I (II, V and VI) in South America.

We planned to undertake sensitivity analyses to determine the effect of restricting the analysis to only: (a) studies with low risk of selection bias (associated with sequence generation or allocation concealment), (b) studies with low risk of performance bias (associated with issues of blinding), and (c) studies with low risk of attrition bias (associated with completeness of data).

Finally, we used the GRADE profiler software package[24–26] in order to assign a level of evidence around the data extracted and to generate pooled estimates and their CIs, and to produce summary of findings tables.

## Results

We identified 803 records through the database search and one additional record by contacting authors. After removing duplicates, 655 records were screened by title and abstract and 22 by full-text. Ultimately, 10 studies (four of them ongoing studies) met our inclusion criteria (Fig 1).

### Included and ongoing studies

We included six completed RCTs[16, 17, 27–30] and identified four ongoing studies[18, 31–33] that met our inclusion criteria(See Table 1 and S3 Text). These studies directly compared fixed vs adjusted doses of BZN. Of the included studies, four were already published[17, 28–30] and details of the remaining two[16, 27] were obtained by personal communication with the authors.

Patients were recruited from Argentina, Bolivia, and Spain (two studies each), and from Brazil, Chile, Mexico, and Guatemala (one study each). Three studies used an adjusted dose [27, 28, 30] and three fixed doses[16, 17, 29]. One study was started before 1997 and used serology and xenodiagnosis[28], those that recruited patients from 1999 to 2018 used PCR for parasitological outcomes. Only one study provided long-term clinical outcomes, including cardiovascular mortality and progression of cardiomyopathy, but for comparison reasons we used a shorter follow-up period as in the majority of studies for non-clinical outcomes[27]. The mean sample size was 330 (minimum 77, maximum 910).

Although the ongoing studies are not included in the evidence synthesis, we have described them in detail (see Table 1) in order to explain how they might be able to answer our main question in the near future. Patients in these studies are being recruited from Argentina, Bolivia, Brazil, Colombia, and Spain. Only one of them will use an adjusted dose[31] and all of them will use PCR to detect presence of parasites.

### Excluded studies

We excluded 12 studies, three of these were duplicate records. The reasons for excluding the other nine[34–42] are described in S3 Text; seven of these were excluded due to a wrong study design.

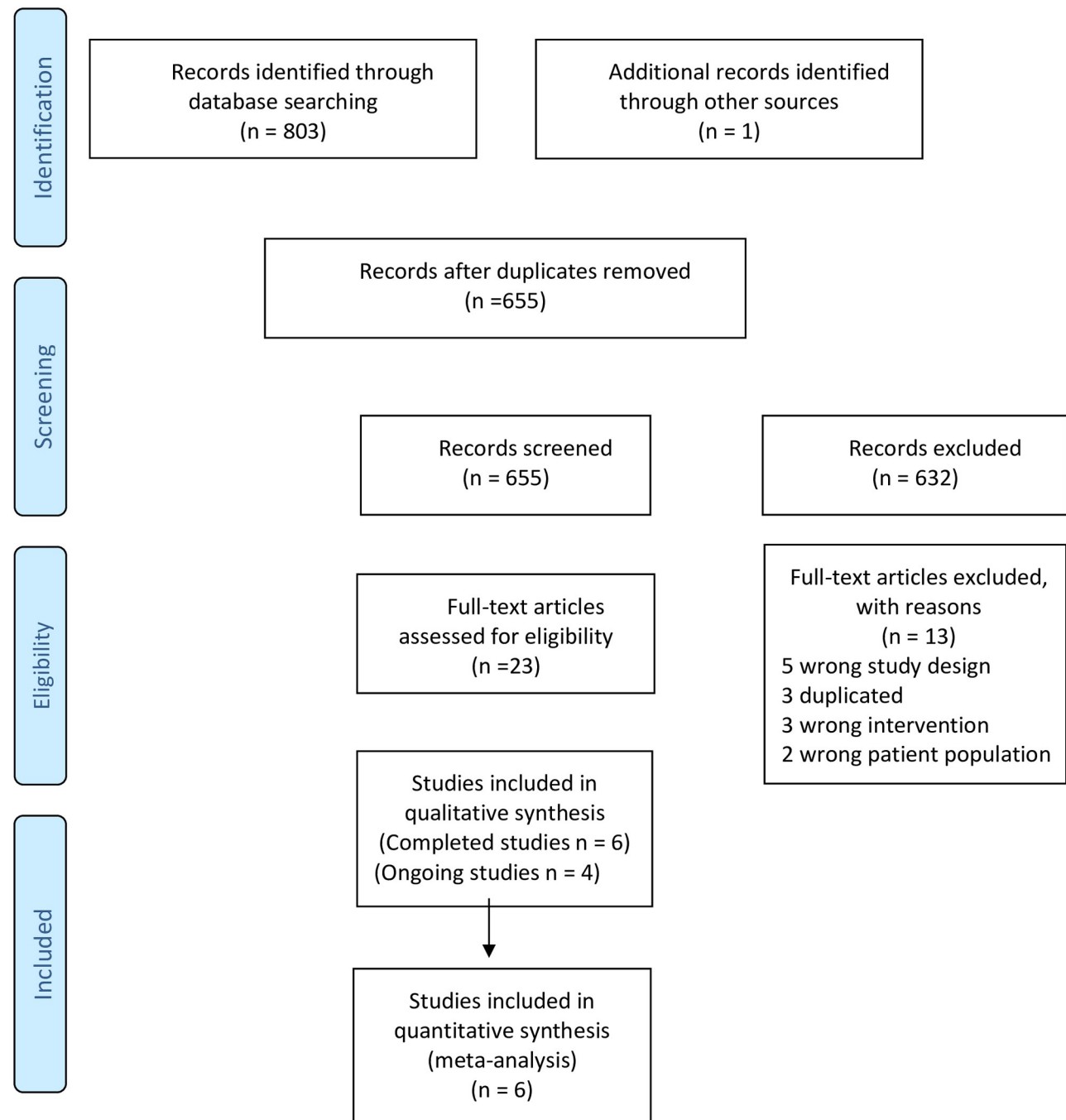

**Fig 1. Study flow diagram. O**: Objective outcomes; **S**: Subjective outcomes.

## Risk of bias in included studies

Only one study was considered of unclear risk of bias for the randomization domains[17]; two studies were considered of high risk for blinding assessment[17, 29] and one study for blinding of participants and personnel[29]; and two studies presented unclear risk for selective reporting[28, 43](see Fig 2, and in S4 Text).

**Table 1. Main characteristics of included and ongoing studies.**

| Short title | PI | Start/end years | N | Countries | Study status | Benznidazole dose | Comparison | Outcomes |
|---|---|---|---|---|---|---|---|---|
| Rodrigues C. 1997[28] | Rodrigues Coura, J. | <1997 | 77 | Brazil | Completed | Adjusted | Nifurtimox/ Placebo | Serology / Xenodiagnoses |
| E1224[30] | Torrico, F. | 2011/2 | 231 | Bolivia | Completed | Adjusted | E1224 / Placebo | PCR |
| CHAGASAZOL [29] | Molina, C. | 2010/1 | 79 | Spain | Completed | Fixed | Posaconazole | PCR |
| STOP-CHAGAS[17] | Morillo, I. | 2011/3 | 120 | Argentina, Chile, México, Guatemala, Spain | Completed | Fixed | Posaconazole / Placebo | PCR |
| BENDITA[16] | Torrico, F. | 2017/8 | 210 | Bolivia | Completed Unpublished | Fixed | E1224 / Placebo | PCR |
| TRAENA[27] | Riarte, A. | 1999/ 2015 | 910 | Argentina | Completed Unpublished | Adjusted | Placebo | PCR Serology Cardiovascular Mortality, Progression |
| BETTY[32] | Buekens, P. | 2019/ | 600 | Argentina | Ongoing | Fixed | Benznidazole 300 mg | PCR |
| CHICAMOCHA 3[31, 47, 48] | Villar, JC. | 2015/ | 500 | Colombia | Ongoing | Adjusted | Nifurtimox/ Placebo | PCR |
| MULTIBENZ [18] | Molina, I. | 2017/ | 240 | Spain, Brazil Argentina, Colombia | Ongoing | Fixed | Benznidazole 150/ 400 mg | PCR |
| TESEO[33] | Almeida, IC. | 2019 | 450 | Bolivia | Ongoing | Fixed | Benznidazole 150/ 300 mg Nifurtimox 240/ 480 mg | RT-PCR |

## Effects of interventions

In considering the main question of this review, we focused the results on comparisons that included both fixed and adjusted doses of BZN, for which we presented GRADE summary of finding tables.

## Benznidazole fixed vs adjusted dose

There was no head-to-head study exploring this comparison but, based on inferences from indirect comparisons of BZN-treated patients versus placebo, using PCR at one year and safety outcomes (see BZN versus placebo below), we did not observe important differences between fixed and adjusted doses (see Table 2). The certainty of evidence for these outcomes was downgraded one or two levels because of indirectness.

## Benznidazole at different fixed doses

We identified the unpublished study BENDITA[16], which did not find important differences in positive PCR between BZN 150 vs 300 for 8 weeks (RR 1.00; IC95% 0.24–4.18), 150 vs 300 for 4 weeks (RR 1.20; IC95% 0.27–5.25), and 150 vs 300 for 2 weeks (RR 0.86; IC95% 0.21, 3.47). The authors also found no differences in AEs for the same comparisons: RR 0.88 (IC95% 0.56, 1.36), RR 1.05 (IC95% 0.65, 1.69), and RR 0.86 (IC 95% 0.49, 1.50) respectively. There were very few serious adverse events (SAEs) and drug discontinuations due to AEs, and no evidence of differences between groups. However, the study design was unpowered to detect differences between arms.

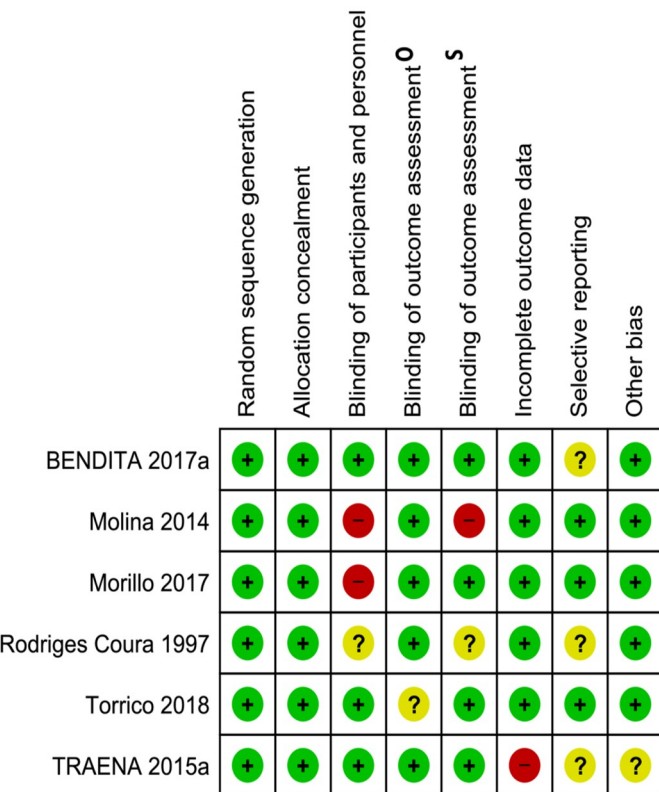

**Fig 2. Risk of bias item for each included study. A**) Random sequence generation (selection bias), (**B**) allocation concealment (selection bias), (**C**) blinding of participants and personnel (performance bias), (**D**) blinding of outcome assessment (detection bias), (**E**) incomplete outcome data (attrition bias), (**F**) selective reporting (reporting bias), (**G**) other bias O: Objective outcomes, S: subjective outcomes.

## Benznidazole versus placebo

Efficacy was determined by positive serology, positive PCR, positive xenodiagnosis, mean reduction of antibody titer, mean reduction in PCR load, and clinical outcomes at the end of follow-up.

Positive serology: we identified one published study[28] without differences between groups (RR 1.00; IC95% 0.93–1.08) and one study[27] that favors treatment with BZN (RR 0.88; IC95% 0.84–0.93), both using an adjusted dose. The pooled RR was 0.94 (IC95% 0.82–1.08) and the certainty of evidence was considered low (see Fig 3). The TRAENA study[27] showed a RR of 0.65 (IC95% 0.58–0.63) at six years.

Positive PCR: both the fixed[17] and adjusted doses[16, 30] of BZN were effective at reducing positive PCR at one year against placebo (RR 0.20; IC95% 0.10–0.30) without differences in subgroups between the fixed dose of 300 mg/day for 14 to 56 days and the adjusted dose: RR 0.12 (IC95% 0.04–0.36) and RR 0.19 (IC95% 0.10–0.37), respectively. The significance of the test for subgroup differences was $Chi^2$ = 0.56, df = 1 (P = 0.45), $I^2$ = 0% (see Fig 4 and Table 3).

Positive xenodiagnosis: only one study provided data about an adjusted dose[28], favoring BZN (RR 0.12, 95% CI 0.04 to 0.36; participants = 60; studies = 1).

Mean reduction in PCR load (GMT one year): Two studies provided data about an adjusted dose[27, 30] and showed no significant difference: (MD -0.48; 95% CI -1.19, 0.23, participants = 480; studies = 2).

**Table 2. Summary of findings: Benznidazole fixed vs adjusted dose.**

| Outcome | Impact | № of participants (studies) | Certainty of the evidence |
|---|---|---|---|
| **Efficacy*** | | | |
| **Positive PCR** | No important difference between fixed and adjusted dose (subgroup differences I² = 0%) | 152 (2 RCTs) | ⊕⊕⊕◯[1] MODERATE |
| **Cardiovascular mortality** | Based only in the surrogate outcome + PCR, differences between groups on critical outcomes are uncertain | 152 (2 RCTs) | ⊕◯◯◯[2] VERY LOW |
| **Progression of cardiomyopathy** | | 152 (2 RCTs) | ⊕◯◯◯[2] VERY LOW |
| **Safety#** | | | |
| **Drug discontinuation** | No important difference between fixed and adjusted dose (Subgroup differences: I² = 0%) | 846 (3 RCTs) | ⊕⊕⊕◯[1] MODERATE |
| **Peripheral neuropathy** | | 769 (2 studies) | ⊕⊕⊕◯[1] MODERATE |
| **Mild rash** | | 769 (2 studies) | ⊕⊕⊕◯[1] MODERATE |
| **Any serious adverse events** | Based only in the surrogate outcome drug discontinuation, differences between groups on this critical outcome is uncertain | 846 (3 RCTs) | ⊕⊕◯◯[3] LOW |

Refer to text for benznidazole versus placebo: Efficacy (PCR at one year*) and Safety#

[1]Downgraded one level due to serious indirectness, since these are inferences from subgroup analysis of comparisons between BZN and placebo.

[2]Downgraded two levels due to due to very serious indirectness (important uncertainty between the surrogate outcome + PCR and the critical outcomes cardiovascular mortality and progression of cardiomyopathy) and methodological limitations to answering this question.

[3]Downgraded one level due to serious indirectness, since there is uncertainty between surrogate outcome drug discontinuation and SAEs.

Mean reduction of antibody titer at one year: only one study (92 participants) provided data about an adjusted dose[30] and showed no difference with placebo (Conventional ELISA: MD 0.01, 95% CI -0.17 to 0.19 and AT CL−ELISA: MD 0.01, 95% CI -0.07 to 0.09) except when using the geometric mean measured by AT CL−ELISA (MD -0.57, 95% CI -1.08 to -0.06).

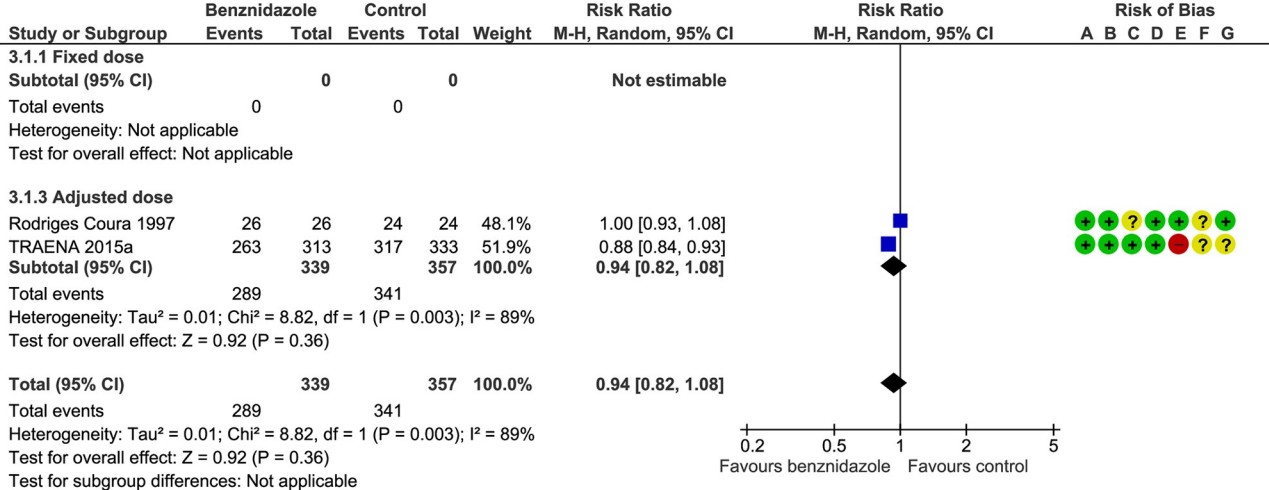

**Fig 3. Positive serology at one year. A)** Random sequence generation (selection bias), (**B**) allocation concealment (selection bias), (**C**) blinding of participants and personnel (performance bias), (**D**) blinding of outcome assessment (detection bias), (**E**) incomplete outcome data (attrition bias), (**F**) selective reporting (reporting bias), (**G**) other bias.

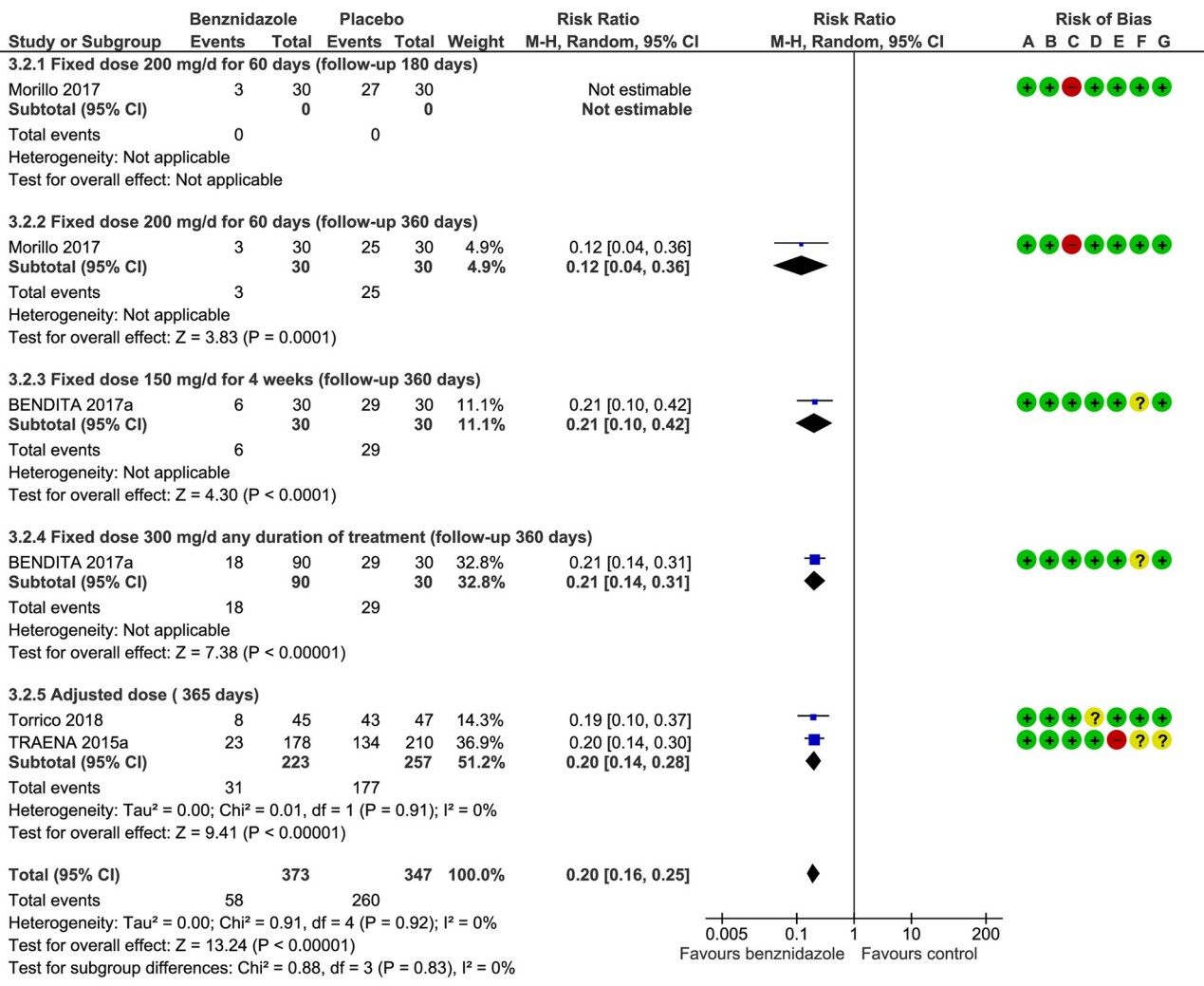

**Fig 4. Positive PCR at one year. A**) Random sequence generation (selection bias), (**B**) allocation concealment (selection bias), (**C**) blinding of participants and personnel (performance bias), (**D**) blinding of outcome assessment (detection bias), (**E**) incomplete outcome data (attrition bias), (**F**) selective reporting (reporting bias), (**G**) other bias.

Clinical outcomes at the end of follow-up: one long-term study (n = 713) that used an adjusted dose of BZN vs. placebo provided data about these outcomes,[27] and showed a RR of 0.89 (95% CI 0.62 to 1.26) for progression of cardiomyopathy, RR 1.18, (95% CI 0.40 to 3.49) for cardiovascular mortality and RR 0.38 (95% CI 0.10 to 1.42) for pacemaker implantation, implantable cardioverters or severe arrhythmia with hemodynamic unbalance, and cardiac failure.

As expected, the frequency of AEs is higher with BZN (see Supporting Information. RevMan, and Raw and analysis data). Only three outcomes presented studies that used fixed or adjusted dose against placebo: drug discontinuation, peripheral neuropathy (considered SAEs), and mild rash (considered non-serious AEs).

Drug discontinuation: both the fixed and adjusted doses showed more drug discontinuation than placebo and we found no subgroup difference: Test for subgroup differences: Chi$^2$ = 0.12, df = 1 (P = 0.73), I$^2$ = 0% 9 (see Fig 5 and Table 3).

**Table 3. Summary of findings: Benznidazole versus placebo.**

| Outcome | Absolute effects (95% CI) | | Relative effect (95% CI) | № of participants (studies) | Certainty of the evidence |
|---|---|---|---|---|---|
| | Risk with placebo: efficacy | Risk with benznidazole | | | |
| **Positive PCR** | 883 per 1.000 | 150 per 1.000 (88 to 265) | RR 0.20 (0.16 to 0.25) | 152 (2 RCTs) | ⊕⊕⊕⊕ HIGH |
| - Fixed dose 300 mg/day for 14 to 56 days | 833 per 1.000 | 100 per 1.000 (33 to 300) | RR 0.12 (0.04 to 0.36) | 60 (1 RCT) | ⊕⊕⊕⊕ HIGH |
| - Adjusted dose | 915 per 1.000 | 174 per 1.000 (91 to 339) | RR 0.19 (0.10 to 0.37) | 92 (1 RCT) | ⊕⊕⊕⊕ HIGH |
| **Drug discontinuation** | 24 per 1.000 | 181 per 1.000 (61 to 533) | RR 7.42 (2.51 to 21.91) | 846 (3 RCTs) | ⊕⊕⊕⊕ HIGH |
| - Fixed dose | 33 per 1.000 | 333 per 1.000 (45 to 1.000) | RR 10.00 (1.36 to 73.33) | 60 (1 RCT) | ⊕⊕⊕⊕ HIGH |
| - Adjusted dose | 24 per 1.000 | 150 per 1.000 (30 to 753) | RR 6.35 (1.27 to 31.86) | 786 (2 RCTs) | ⊕⊕⊕⊕ HIGH |
| **Peripheral neuropathy** | 2 per 1000 | 10 per 1000 (2 to 47) | RR 4.27 (0.94 to 19.40) | 919 (3 RCTs) | ⊕⊕⊕⊕ HIGH |
| - Fixed dose | 0 per 1000 | 0 per 1000 (0 to 0) | RR 1.52 (0.16 to 14.32) | 210 (2 RCTs) | ⊕⊕⊕⊕ HIGH |
| - Adjusted dose | 3 per 1000 | 28 per 1000 (4 to 221) | RR 10.14 (1.31 to 78.81) | 709 (1 RCT) | ⊕⊕⊕⊕ HIGH |
| **Mild rash** | 67 per 1000 | 357 per 1000 (246 to 520) | RR 5.32 (3.66 to 7.74) | 919 (3 RCTs) | ⊕⊕⊕⊕ HIGH |
| - Fixed dose | 50 per 1000 | 312 per 1000 (111 to 876) | RR 6.24 (2.22 to 17.52) | 210 (2 RCTs) | ⊕⊕⊕⊕ HIGH |
| - Adjusted dose | 70 per 1000 | 363 per 1000 (243 to 544) | RR 5.19 (3.47 to 7.77) | 709 (1 RCT) | ⊕⊕⊕⊕ HIGH |

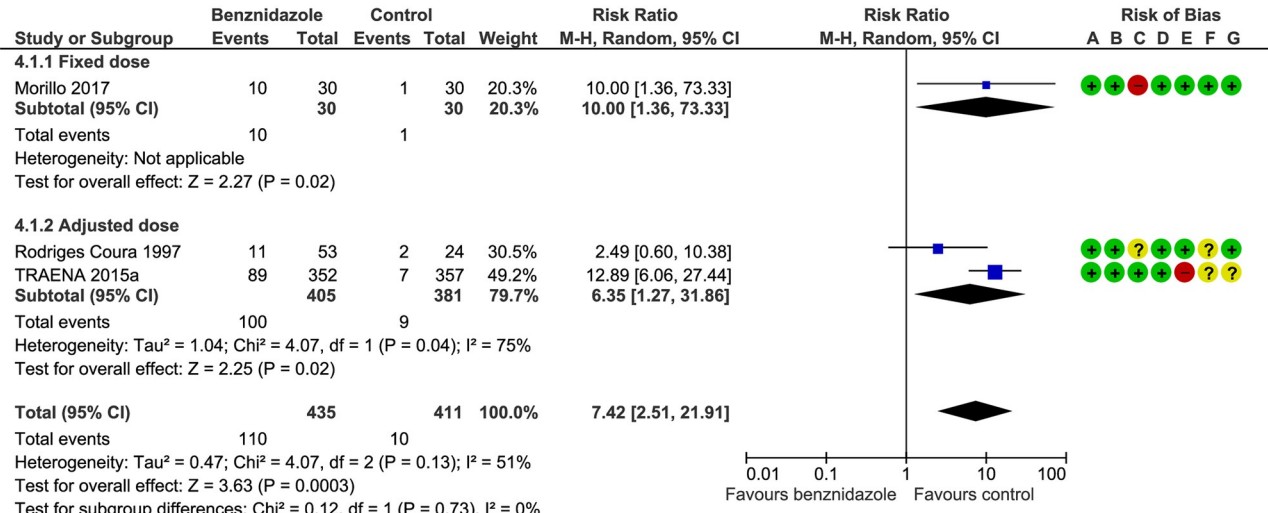

**Fig 5. Drug discontinuation. A**) Random sequence generation (selection bias), (**B**) allocation concealment (selection bias), (**C**) blinding of participants and personnel (performance bias), (**D**) blinding of outcome assessment (detection bias), (**E**) incomplete outcome data (attrition bias), (**F**) selective reporting (reporting bias), (**G**) other bias.

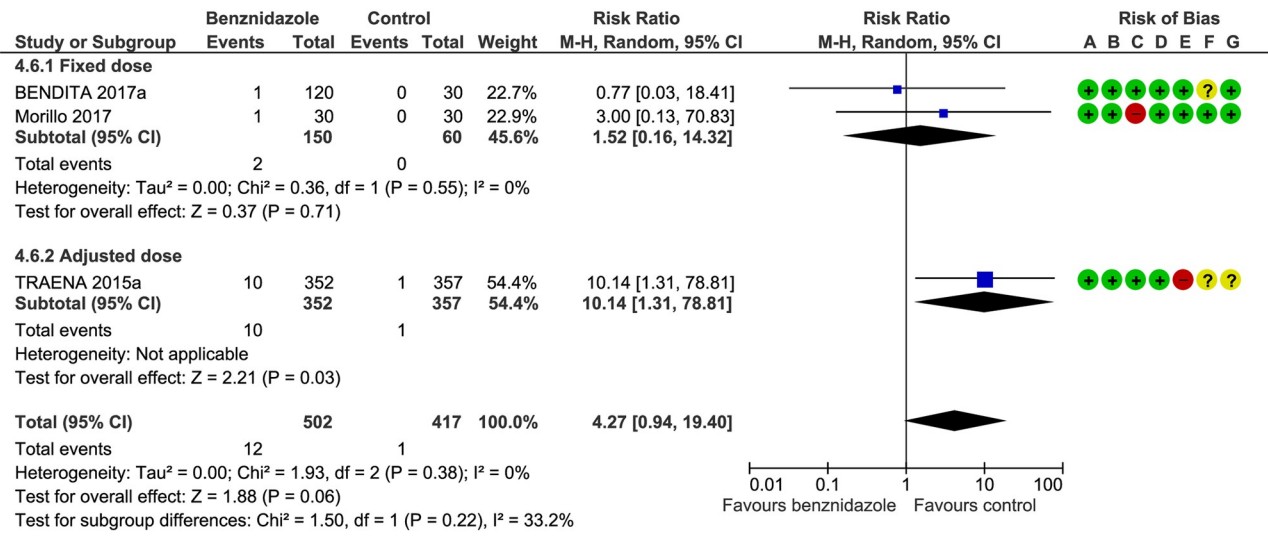

**Fig 6. Peripheral neuropathy. A**) Random sequence generation (selection bias), (**B**) allocation concealment (selection bias), (**C**) blinding of participants and personnel (performance bias), (**D**) blinding of outcome assessment (detection bias), (**E**) incomplete outcome data (attrition bias), (**F**) selective reporting (reporting bias), (**G**) other bias.

Peripheral neuropathy: we found no subgroup differences: Test for subgroup differences: $Chi^2 = 1.50$, df = 1 (P = 0.22), $I^2 = 33.2\%$. See Fig 6 and Table 3.

Mild rash: we found no subgroup difference: Test for subgroup differences: $Chi^2 = 0.14$, df = 1 (P = 0.93), $I^2 = 0\%$) (see Fig 7 and Table 3).

## Benznidazole versus posaconazole

We identified two studies that used fixed-dose BZN versus posaconazole[17, 29] and considering the longest follow-up period, both favored BZN (RR 0.24, 95% CI 0.06 to 0.93;

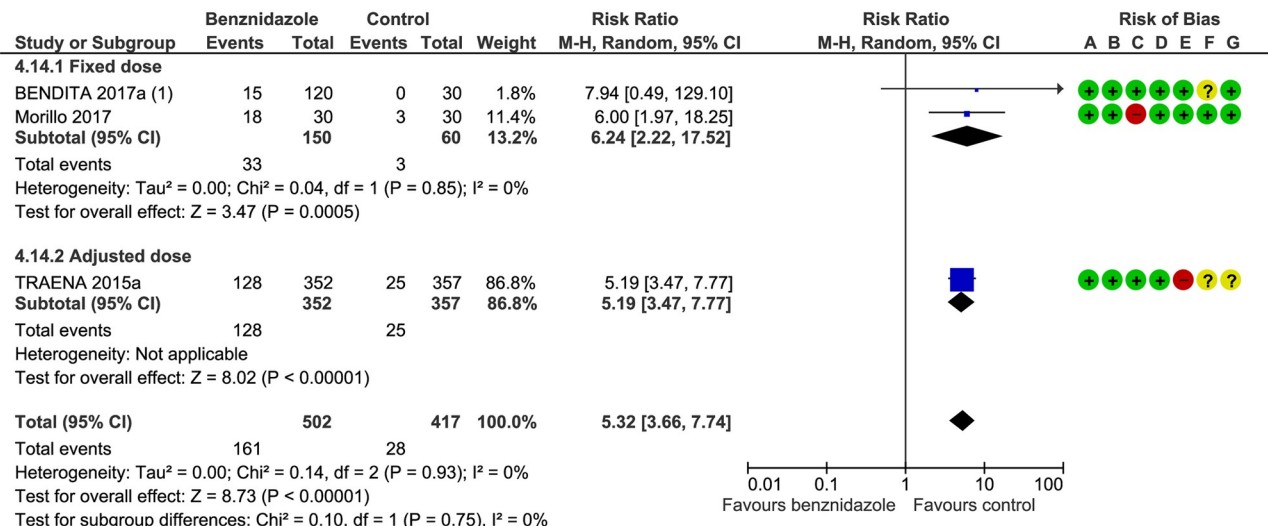

**Fig 7. Mild rash.** (**1**) Bendita 20117 (Maculo-papular rash + papular rash + erythema. **A**) Random sequence generation (selection bias), (**B**) allocation concealment (selection bias), (**C**) blinding of participants and personnel (performance bias), (**D**) blinding of outcome assessment (detection bias), (**E**) incomplete outcome data (attrition bias), (**F**) selective reporting (reporting bias), (**G**) other bias.

participants = 112; studies = 2; $I^2$ = 81%). Test for subgroup differences by follow-up: $Chi^2$ = 4.17, df = 1 (P = 0.04), $I^2$ = 76.0% due to higher effect in Morillo 2017 (follow-up 360 days)[17] than in Molina 2014 (follow-up 280 days)[29].

## Subgroup analysis and sensitivity analysis

Due to the low number of included studies it was not possible to conduct the planned subgroup analyses, except for the fixed and adjusted doses related to the question of our review.

For the same reason, it was not possible to conduct the planned sensitivity analyses, restricting the analysis to only studies with low risk of selection bias (associated with sequence generation or allocation concealment). Morillo et al.[17] (unclear risk of bias for this domain) was the only fixed-dose study included in the subgroup comparison between fixed and adjusted doses of BZN.

All studies included in the subgroup comparison between fixed and adjusted doses of BZN were of low risk of performance bias (associated with issues of blinding) and low risk of attrition bias (associated with completeness of data) except for TRAENA[27]. We found consistency of results applying both fixed-effect and random-effects models and also using OR and RD.

## Discussion

The only drugs with proven efficacy against Chagas disease are BZN and nifurtimox. BZN is used in children and adults and is registered for use in adjusted-dose schemes. However, some investigators have proposed the more flexible use of fixed dosing regardless of body weight. After searching all completed and ongoing RCTs involving BZN at any dose, we found no direct comparison between fixed and adjusted doses of BZN.

We only found one efficacy outcome (positive PCR) and three safety outcomes (drug discontinuation, peripheral neuropathy, and mild rash) that allowed the subgroup comparisons between fixed and adjusted doses of BZN [16, 17, 27, 28, 30] The low or null $I^2$ for all these subgroup comparisons suggest no important clinical, methodological, or statistical differences in the observed effects by type of dosing. Since these are inferences from indirect comparisons of BZN-treated patients versus placebo, the certainty of evidence for these outcomes was consequently downgraded one or two levels because of indirectness.

We found a high certainty of evidence for the direct comparisons between BZN versus placebo for the efficacy outcome (positive PCR) and for the three safety outcomes (drug discontinuation, peripheral neuropathy and mild rash). However, these four outcomes are considered as moderate certainty of the evidence for the comparison fixed vs adjusted dose of benznidazole after downgrading one level due to indirectness (indirect comparisons). The certainty of the evidence was considered to be very low for cardiovascular mortality and progression of cardiomyopathy due to significant uncertainty between the surrogate positive PCR outcome and these critical outcomes. The certainty of the evidence was considered low for any SAE due to uncertainty between this outcome and the surrogate outcome drug discontinuation. See Table 2 (based on indirect comparisons) and Table 3 (for direct comparisons between BZN and placebo by type of dose).

We found six related systematic reviews that showed similar results in terms of the effect of interventions against placebo, however, none of them addressed our question concerning the comparison of fixed and adjusted doses of BZN [38, 39, 42, 44–46]. Observational studies suggest that treatment could be better that no treatment even in the early phases of CCC[46]. Unfortunately, non-RCT studies used adjusted doses of BZN, not allowing the assessment of subgroup analysis by BZN scheme. Unlike other reviews, we only included RCTs to reduce the

risk of bias, but, as was the case for the previous studies, we had to deal with differences in the populations studied, follow-up periods, diagnostic techniques, and sample size.

Demonstration that a fixed dose of BZN has a similar profile in relation to safety and efficacy would allow a review of the current guidelines and the recommendation of fixed doses, eliminating one barrier to treatment management, and improving compliance by patients and health workers.

The strengths of this systematic review of RCTs include the registration of its protocol, the complete literature search, the rigorous Cochrane methods used, the participation of most principal investigators of the RCTs included, and the inclusion of valuable unpublished data. All these factors make our study the most complete evidence synthesis currently available that addresses the comparative efficacy and safety of adjusted-dose BZN for *T. cruzi* seropositive adults mainly without CCC.

The population included in the trials is representative of the population of adults with chronic Chagas disease without cardiomyopathy, however important limitations need to be mentioned. First of all, the absence of direct comparisons between fixed and adjusted doses of BNZ, and the assessments of critical outcomes should be noticed. Additionally, the paucity of studies prevented us from performing our planned subgroup and sensitivity analyses. The unpublished TRAENA study[27] was the only one that succeeded in assessing long-term clinical outcomes.

Individual patient data (IPD) meta-analysis could address these issues, for example by including cumulative dose assessments. Moreover, network IPD meta-analysis could formally enhance indirect comparisons.

## Conclusion

Based on a low to very low certainty of evidence for critical clinical outcomes and a moderate certainty of evidence for important outcomes, fixed and adjusted doses of BZN might be considered equivalent in terms of efficacy and safety.

An IPD meta-analysis would allow us to conduct the planned subgroup analysis and meta-regressions, but given the absence of a direct comparison between fixed and adjusted doses of BZN, the only approach to gain in certainty of evidence to address the objective of the review would be an IPD network-meta-analysis—an approach which our research group is currently following.

## Supporting information

**S1 Text. Search strategy.**
(DOCX)

**S2 Text. Risk of bias assessment tool.**
(DOCX)

**S3 Text. Detailed description of included and ongoing studies and Excluded studies and reasons for exclusion.**
(DOCX)

**S4 Text. Support for judgement of included studies by risk of bias item and Risk of bias graph across all included studies.**
(DOCX)

**S5 Text. RevMan file.**
(HTM)

**S1 PRISMA Checklist.**
(DOCX)

**S1 Data. Raw and analysis data.**
(CSV)

## Acknowledgments

We want to acknowledge Daniel Comandé for his contribution to the search strategy.

## Author Contributions

**Conceptualization:** Agustín Ciapponi, Fabiana Barreira, Lucas Perelli, Ariel Bardach, Joaquim Gascón, Israel Molina, Carlos Morillo, Nilda Prado, Adelina Riarte, Faustino Torrico, Isabela Ribeiro, Juan Carlos Villar, Sergio Sosa-Estani.

**Data curation:** Agustín Ciapponi, Lucas Perelli, Joaquim Gascón, Israel Molina, Carlos Morillo, Nilda Prado, Adelina Riarte, Isabela Ribeiro, Juan Carlos Villar.

**Formal analysis:** Agustín Ciapponi, Lucas Perelli, Ariel Bardach.

**Investigation:** Agustín Ciapponi, Lucas Perelli, Ariel Bardach.

**Methodology:** Agustín Ciapponi, Lucas Perelli, Ariel Bardach.

**Project administration:** Agustín Ciapponi, Fabiana Barreira, Lucas Perelli, Ariel Bardach, Sergio Sosa-Estani.

**Resources:** Agustín Ciapponi.

**Software:** Agustín Ciapponi.

**Supervision:** Agustín Ciapponi, Ariel Bardach, Sergio Sosa-Estani.

**Validation:** Agustín Ciapponi, Ariel Bardach, Sergio Sosa-Estani.

**Visualization:** Agustín Ciapponi.

**Writing – original draft:** Agustín Ciapponi, Lucas Perelli, Ariel Bardach.

**Writing – review & editing:** Agustín Ciapponi, Fabiana Barreira, Lucas Perelli, Ariel Bardach, Joaquim Gascón, Israel Molina, Carlos Morillo, Nilda Prado, Adelina Riarte, Faustino Torrico, Isabela Ribeiro, Juan Carlos Villar, Sergio Sosa-Estani.

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
