## [Editor Report · Decision Letter 0]

30 Mar 2020

Dear Dr. Ciapponi,

Thank you very much for submitting your manuscript "Fixed vs adjusted-dose benznidazole for adults with chronic Chagas disease without cardiomyopathy: A systematic review and meta-analysis" for consideration at PLOS Neglected Tropical Diseases. As with all papers reviewed by the journal, your manuscript was reviewed by members of the editorial board, which appreciated the attention to an important topic. We will likely consider your manuscript for publication, providing that you modify the manuscript according to the editorial review recommendations. 

We noted several problems with manuscript formatting as detailed bellow. Please, proceed to fix the texts, formatting and graphs and return the manuscript for proper editorial flow as soon as possible.

Please prepare and submit your revised manuscript within 10 days. If you anticipate any delay, please let us know the expected resubmission date by replying to this email.  

Sincerely,

Helton da Costa Santiago, M.D., Ph.D

Associate Editor

Alain Debrabant, Ph.D

Deputy Editor

Before we send the manuscript out for peer revision, we would like to request the authors to review the text and to fix the figure 3. The text in lines 447- refers to serology outcome evaluation, but refers to figure 4, which show PCR data whereas supposedly it should refer to figure 3, which is very confusing. The title of what seems to be figure 3 is the same as figure 4, but the data apparently refers to a different data set, as it is supposed to be. However, it is not clear by looking at the figure what is the data being displayed.

In addition, we would like to request the authors to follow authors guidelines in formatting the manuscript. Although, Plos NTDs might be very flexible in a first submission regarding to formatting, there are several problems with the manuscript that makes the reading annoying:

1- References are repeated in between methods sessions and in the end of the manuscript

2- All tables are repeated in the methods session and in the end of the manuscript

3- Legend of table 2 is misplaced between the discussion session.

Please, proceed to fix the texts, formatting and graphs and return the manuscript for proper editorial flow as soon as possible.
---

## [Decision Letter · Decision Letter 1]

19 Jun 2020

Dear Dr. Ciapponi,

Thank you very much for submitting your manuscript "Fixed vs adjusted-dose benznidazole for adults with chronic Chagas disease without cardiomyopathy: A systematic review and meta-analysis" for consideration at PLOS Neglected Tropical Diseases. As with all papers reviewed by the journal, your manuscript was reviewed by members of the editorial board and by several independent reviewers. The reviewers appreciated the attention to an important topic. Based on the reviews, we are likely to accept this manuscript for publication, providing that you modify the manuscript according to the review recommendations. 

The manuscript was evaluated by three experts in field. One reviewer rejected the manuscript based on the limited number papers found to be included in the meta-analysis. The editors agree that the limited number of manuscript imposes a limitation, but we acknowledge that the authors are aware of the limitations, which are disclosed in the discussion. We are requesting the authors to perform minor revision as they find appropriate on the suggestions offered by reviewer #3.

Sincerely,

Helton da Costa Santiago, M.D., Ph.D

Associate Editor

Alain Debrabant, Ph.D.

Deputy Editor

The manuscript was evaluated by three experts in field. One reviewer rejected the manuscript based on the limited number papers found to be included in the meta-analysis. The editors agree that the limited number of manuscript imposes a limitation, but we acknowledge that the authors are aware of the limitations, which are disclosed in the discussion. We are requesting the authors to perform minor revision as they find appropriate on the suggestions offered by reviewer #3.

Reviewer's Responses to Questions

**Key Review Criteria Required for Acceptance?**

**Methods**

-Are the objectives of the study clearly articulated with a clear testable hypothesis stated?

-Is the study design appropriate to address the stated objectives?

-Is the population clearly described and appropriate for the hypothesis being tested?

-Is the sample size sufficient to ensure adequate power to address the hypothesis being tested?

-Were correct statistical analysis used to support conclusions?

-Are there concerns about ethical or regulatory requirements being met?

Reviewer #1: While the authors present an interesting premise this study has some issues. 

They propose to determine the comparative safety and efficacy of a fixed dose of benznidazole (BZN) with an adjusted dose for the treatment of T. cruzi seropositive adults without cardiomyopathy through a systematic review and a meta-analysis.

After the search they found ten studies that meet the inclusion criteria. However, four of them are ongoing and two are unpublished. So, these studies should not be included in the analysis. They did not find any study comparing a fixed dose X an adjusted dose of BZN. All these together make difficult to perform a meta-analysis study and address this important question.

So, this manuscritp should be rejected

Reviewer #2: In my opinion, after authors revision, the study is clearly articulated with hypothesis, the design is appropriate to the stated objectives. And, despite of sample size is very low the conclusion is suitable from found results.

Reviewer #3: The study objective was properly stated. Method and adequate analysis were done.

**Results**

-Does the analysis presented match the analysis plan?

-Are the results clearly and completely presented?

-Are the figures (Tables, Images) of sufficient quality for clarity?

Reviewer #1: Figure 3 is missing

Reviewer #2: The results are presented clearly and they are completely matched to analysis plan.

Reviewer #3: The results met planned analysis and were presented clearly. Study importance as well as limitations were adequately described.

**Conclusions**

-Are the conclusions supported by the data presented?

-Are the limitations of analysis clearly described?

-Do the authors discuss how these data can be helpful to advance our understanding of the topic under study?

-Is public health relevance addressed?

Reviewer #1: Conclusions can not be supported by the meta-analysis performed. Among the ten studies selected two of them have not yet been published and four were ongoing studies and should no be included in the analysis.

Reviewer #2: The conclusion is supported by the data presented, the limitations are presented and clearly described.

Reviewer #3: The conclusions provided are in accord with results presented and data analysed in the study.

**Editorial and Data Presentation Modifications?**

Reviewer #1: NO

Reviewer #2: There is no necessity of data modification, after authors revision.

Reviewer #3: (No Response)

**Summary and General Comments**

Reviewer #1: (No Response)

Reviewer #2: Despite of small number of eligible studies the paper is important because direction of specific Chagas disease treatment is not is not clear yet.

Reviewer #3: The study evaluated and reviewed available literature to understand differential in treatment dosage for chagas disease. The result of the meta-analysis provide useful and contribute to existing knowledge about the disease and more importantly, serve as a background on which a ore significant experimental study can be conducted.

PLOS authors have the option to publish the peer review history of their article (what does this mean?). If published, this will include your full peer review and any attached files.

Reviewer #1: No

Reviewer #2: No

Reviewer #3: Yes: Adebiyi Adeniran
---

## [Editor Report · Decision Letter 2]

26 Jun 2020

Dear Dr. Ciapponi,

We are pleased to inform you that your manuscript 'Fixed vs adjusted-dose benznidazole for adults with chronic Chagas disease without cardiomyopathy: A systematic review and meta-analysis' has been provisionally accepted for publication in PLOS Neglected Tropical Diseases.

Best regards,

Helton da Costa Santiago, M.D., Ph.D

Associate Editor

Alain Debrabant, Ph.D.

Deputy Editor

---

## [Editor Report · Acceptance letter]

7 Aug 2020

Dear Dr. Ciapponi,

We are delighted to inform you that your manuscript, "Fixed vs adjusted-dose benznidazole for adults with chronic Chagas disease without cardiomyopathy: A systematic review and meta-analysis," has been formally accepted for publication in PLOS Neglected Tropical Diseases.

Best regards,

Shaden Kamhawi

co-Editor-in-Chief

Paul Brindley

co-Editor-in-Chief
